# The Important Role of Membrane Fluidity on the Lytic Mechanism of the α-Pore-Forming Toxin Sticholysin I

**DOI:** 10.3390/toxins15010080

**Published:** 2023-01-16

**Authors:** Lohans Pedrera, Uris Ros, Maria Laura Fanani, María E. Lanio, Richard M. Epand, Ana J. García-Sáez, Carlos Álvarez

**Affiliations:** 1Centro de Estudio de Proteínas, Facultad de Biología, Universidad de La Habana, La Habana CP 10400, Cuba; 2Institute for Genetics and CECAD Cluster of Excellence, University of Cologne, Joseph-Stelzmann-Straße 26, 50931 Cologne, Germany; 3Departamento de Química Biológica Ranwel Caputto, Facultad de Ciencias Químicas, Universidad Nacional de Córdoba, Córdoba X5000HUA, Argentina; 4Centro de Investigaciones en Química Biológica de Córdoba (CIQUIBIC), Facultad de Ciencias Químicas-CONICET, Córdoba X5000HUA, Argentina; 5Department of Biochemistry and Biomedical Sciences, Faculty of Health Sciences, McMaster University, Hamilton, ON L8S 4K1, Canada

**Keywords:** actinoporins, pore-forming toxins, membrane permeabilization, membrane fluidity, lipid phase-coexistence

## Abstract

Actinoporins have emerged as archetypal α-pore-forming toxins (PFTs) that promote the formation of pores in membranes upon oligomerization and insertion of an α-helix pore-forming domain in the bilayer. These proteins have been used as active components of immunotoxins, therefore, understanding their lytic mechanism is crucial for developing this and other applications. However, the mechanism of how the biophysical properties of the membrane modulate the properties of pores generated by actinoporins remains unclear. Here we studied the effect of membrane fluidity on the permeabilizing activity of sticholysin I (St I), a toxin that belongs to the actinoporins family of α-PFTs. To modulate membrane fluidity we used vesicles made of an equimolar mixture of phosphatidylcholine (PC) and egg sphingomyelin (eggSM), in which PC contained fatty acids of different acyl chain lengths and degrees of unsaturation. Our detailed single-vesicle analysis revealed that when membrane fluidity is high, most of the vesicles are partially permeabilized in a *graded* manner. In contrast, more rigid membranes can be either completely permeabilized or not, indicating an *all-or-none* mechanism. Altogether, our results reveal that St I pores can be heterogeneous in size and stability, and that these properties depend on the fluid state of the lipid bilayer. We propose that membrane fluidity at different regions of cellular membranes is a key factor to modulate the activity of the actinoporins, which has implications for the design of different therapeutic strategies based on their lytic action.

## 1. Introduction

Pore-forming toxins (PFTs) are proteins able to produce holes in their target cell membranes altering their permeability. Sticholysin I (St I) is a potent PFT produced by the sea anemone *Stichodactyla helianthus* belonging to the actinoporin protein family, a unique class of PFTs found in sea anemones [1]. Actinoporins are basic proteins made of a single polypeptide chain, with molecular weight of around 20 kDa and high affinity for SM-containing membranes provided this SM is in a fluid environment [2]. Actinoporins have attracted scientific attention as model systems to understand the basic molecular mechanisms of protein–membrane insertion and permeabilization and to develop biomedical applications. Therefore, understanding how biophysical membrane properties modulate the pore-forming activity of actinoporins has biological and medical relevance.

A distinctive structural feature of actinoporins is the presence of an aromatic amino acid cluster exposed to the aqueous medium and a phosphorylcholine (POC) binding site involved in the interaction with lipid membranes [3]. Furthermore, the structural characterization of fragaceatoxin C, an actinoporin from *Actinia fragacea* revealed the existence of five sites for membrane lipid binding: two primary sites for phospholipids containing POC at the polar headgroup, two other low-affinity sites, or perhaps high-affinity binding sites, for lipids with headgroups other than POC and a lipid site connecting two adjacent monomers in the oligomeric structure [4]. These findings suggest the relevance of the lipids, in particular SM, not just as membrane receptors but also as structural elements or cofactors of the pore.

Current models of pore-formation by actinoporins include binding to lipid membrane, oligomerization, N-terminal insertion and final pore-formation [5,6,7,8]. It is very likely that during pore structuration each pore passes through different states via the incorporation of a growing number of α-helical protein sequences and lipids [5,7], which can explain the identification of pores with different stoichiometry [9]. Over the years, different groups have studied the mechanism of pore formation by actinoporins using minimalist model membrane systems of different lipid compositions. These studies showed that membrane binding and pore formation by actinoporins is strongly affected by different membrane biophysical properties such as membrane fluidity [10,11], bilayer thickness [10], lateral phase coexistence [10,12,13,14] and line tension [15,16]. However, until now it is not clear how membrane biophysical properties impact the stability and the size of the pores formed by actinoporins.

Here, we investigated the effect of membrane fluidity on binding and pore-forming activity of St I, combining population analysis of small unilamellar vesicles (SUVs) and single-vesicle analysis using giant unilamellar vesicles (GUVs). We used lipid vesicles of different compositions, taking as the basis the equimolar phosphatidylcholine (PC) and egg sphingomyelin (eggSM) mixture in which fluidity was modulated by changing the acyl chain length and unsaturation of the PC. Our results reinforce the notion that fluid vesicles characterized by weaker lipid cohesion provide a more suitable environment for St I lipid binding and subsequent pore formation. Our results indicate that fluid membrane vesicles are partially permeabilized by St I following a *graded* mechanism, whereas in a more ordered environment the pores reach a more stable open-state typical of an *all-or-none* mode of action. Moreover, we demonstrated that the pore formed by St I is tunable in size and can be modulated by membrane properties. Based on this observation, we proposed that the pores formed by actinoporins in cellular membranes can be modulated by the fluidity of the different regions of the membrane.

## 2. Results

### 2.1. Binding to Lipid Membranes and Pore-Forming Activity of St I Is Optimal at an Equimolar Mixture of POPC:eggSM

To identify the optimal proportion of SM and PC to study the effect of membrane fluidity in the mechanism of action of St I, we studied its binding and pore-forming activity in SUVs made of a broad range of 1-palmitoyl-2-oleoyl phosphatidylcholine (POPC):eggSM ratios (Figure 1A). Binding and accommodation efficiency of St I to the vesicles was determined by following the changes in protein Trp fluorescence emission spectra elicited by vesicles addition as the result of the change in the polarity of Trp environment (Figure 1B,C), whereas the permeabilizing activity was followed by a carboxyfluorescein (CF) release assay (Figure 1D,E). 

The addition of SUVs to St I triggered a progressive increase in fluorescence emission intensity, and a blue shift of maximum fluorescence emission which became stabilized at the highest lipid/St I molar ratios (Figure 1B and Table 1). Vesicles containing a low proportion (15 mol%) or lacking eggSM showed lower values of F_SUVs_/F_0_ and Δλ as well as higher values of C_50%_ in comparison with vesicles containing between 30 and 70% eggSM, suggesting that the series of events comprising binding and oligomer accommodation of St I in liposomes increased with the presence of eggSM (Figure 1B and Table 1). To further characterize these St I processes in POPC:eggSM vesicles, we measured the accessibility of Trp to the water-soluble quencher acrylamide under saturation conditions for all compositions (lipid/St I molar ratio ~ 200) (Figure 1C). The quenching efficiency estimated by the parameter K_SV_ was similar and lower in vesicles containing between 30 and 70 mol% eggSM than for vesicles with a low proportion of eggSM (15 mol%) or pure POPC (Table 1). This indicates that St I has less exposition of its Trp residues to the aqueous medium in the presence of vesicles with eggSM ≥ 30 mol%, and hence, is more inserted in the membrane in comparison to vesicles with low proportion of eggSM (15 mol%) or pure POPC (Table 1).

St I did not exert permeabilizing activity in SUVs exclusively formed by POPC even at high toxin concentrations in correspondence with the low binding extent of the toxin to vesicles of this composition (Figure 1D). In contrast, the pore-forming capacity of St I was maximum at the equimolar proportion of POPC:eggSM and diminished significantly when one of the two phospholipids predominated (Figure 1D,E). Collectively, our experiments show that the binding process and accommodation of St I leading to pore-formation of St I in SUVs of POPC and eggSM is maximum in the equimolar mixture of both phospholipids and decreases when one of the two components prevails.

### 2.2. Membrane Fluidity Favors the Binding and Permeabilizing Activity of St I in Lipid Vesicles

Given the fact that the process leading to pore-formation by St I was higher in POPC:eggSM equimolar proportion, we prepared vesicles containing PC with different acyl chain lengths and unsaturation to systematically investigate the role of membrane fluidity in St I activity. To this end, we selected PCs with symmetric saturated acyl chains of 14 (DMPC), 16 (DPPC), 18 (DSPC), and 20 (DAPC) carbon atoms and the unsaturated 18:1 acyl chain (DOPC) (Figure 2A). We did not observe that the nature of the PCs affected the size of the SUVs in a remarkable way as shown by dynamic light scattering (Appendix A).

The fluidity of SUVs was characterized by steady-state fluorescence measurements resulting from the ability of the fluorescent dye pyrene to dimerize leading to excimer formation, while the lipid phase state of membranes was measured by differential scanning calorimetry (DSC). In gel phase (Lβ) membranes, where the lipid environment restricts the lateral movement of pyrene, fluorescence emission originates primarily from monomers with maximum emission at 375 nm (F_monomer_) [17] (Appendix A). In contrast, in fluid membranes pyrene monomers associate with higher probability resulting in an increase in the fluorescence of the excimeric species at 470 nm (F_excimer_) [17]. Based on our data, we grouped the vesicles into two well-defined families: liposomes with high fluidity containing the shorter (DMPC) or unsaturated PC (DOPC), and vesicles with lower fluidity containing saturated PC between 16 and 20 C atoms (Appendix A).

The determination of the thermotropic transition of the vesicles by DSC showed that the longer the PC acyl chain, the higher L_β_/liquid crystalline (L_α_) phase transition temperature (Tm) (Appendix A). Interestingly, vesicles of DAPC:eggSM showed two Tm, typical of a lipid membrane with phase coexistence: one DAPC-enriched (~53 °C) and the other eggSM-enriched (41 °C). To compare the physical state of different vesicles in our functional assays, we analyzed the phase state of the liposomes at room temperature (23 ± 2 °C). At this temperature, vesicles of DOPC:eggSM whose transition phase starts at 10 °C and finish at 36 °C, and DMPC:eggSM whose calorimetric peak is between 18 and 40 °C, are both in the temperature range with Lβ/Lα phase coexistence. In contrast, the rest of the vesicles are below the transition calorimetric range at room temperature and therefore in Lβ phase (DPPC 33–43 °C; DSPC 33.5–60 °C; DAPC 35–57 °C) (Appendix A). The biophysical characterization of the vesicles showed that the increase in the acyl chain length turned the vesicles less fluid while the introduction of one unsaturation in both chains increased fluidity.

Regardless of the type of PC, St I associated to vesicles in a similar extent, as evidenced by similar values of F_SUVs_/F_0 max_ and K_SV_, obtained at high lipid/protein molar ratios (Figure 2B,C, Table 2). However, the parameter L_50%_ reflected that St I and bound oligomers had around a 7-fold higher affinity for eggSM vesicles containing DOPC in comparison with liposomes containing saturated PCs between 14 and 18C, with hardly any difference between the latter. In contrast, St I and other molecular aggregates in membrane showed the lowest affinity for eggSM vesicles containing DAPC (Figure 2B,C, Table 2). Moreover, the pore-forming activity of St I was higher in eggSM vesicles containing DMPC or DOPC without differences among them and decreased significantly with the increase in the acyl chain length (Figure 2D,E and Appendix A). Interestingly, when St I pore-forming capacity was compared under binding saturation conditions (lipid/St I molar ratio ~200:1, Figure 2B), DMPC:eggSM or DOPC:eggSM vesicles were more susceptible to St I action than the eggSM vesicles containing saturated PC between 16 and 20 C atoms (Figure 2D). Under these conditions, all the protein is bound, therefore the total protein/liposome ratio is very similar (Figure 2B). For this reason, it is possible to deduce that the higher permeabilizing activity of St I in vesicles containing DMPC or DOPC is due to differences in the pore formation stages subsequent to the binding step (Figure 2B). 

To get insight into the effect of membrane fluidity on the mode of action of St I, we directly visualized its binding and pore-forming capacity in GUVs using confocal microscopy. For this, we selected the lipid mixtures with similar high fluidity (DMPC or DOPC) and with low fluidity (DAPC) (Appendix A). Both group of SUVs differed extremely in their permeabilizing behaviors (Figure 2E). Moreover, we used a fluorescently labeled St I (St I 125C-Alexa) to visualize binding of the toxin to the GUVs and followed vesicle permeabilization by the influx of free Alexa 488. Before its use, we checked that the labelling process did not affect the pore-forming activity of the mutant St I 125C (Appendix A). The binding and extent of permeabilization by St I were higher in DOPC:eggSM and DMPC:eggSM GUVs than in DAPC-containing vesicles (Figure 2F–H). Moreover, the GUVs system allowed detecting differences between the DMPC and DOPC containing vesicles that were not evident with SUVs, being the DOPC containing GUVs more sensitive to toxin binding and permeabilization. This difference was possibly due to a higher working temperature in GUVs closer to the DMPC:eggSM Tm temperature. Altogether, our results suggest that in both SUVs and GUVs systems the fluid vesicles were more sensitive to the toxin action of St I (Figure 2 and Appendix A).

### 2.3. Membrane Fluidity Regulates the Mechanism of Membrane Permeabilization by St I

The detailed characterization of the permeabilization mechanism in bulk vesicles experiments is masked by the heterogeneity in which vesicles in the SUVs population undergo permeabilization upon toxin exposition. To address this issue, we followed the increase in the Alexa 488 free dye fluorescence inside the vesicles at the single-vesicle level by time-lapse confocal microscopy (Figure 3A–D). The permeabilization of the vesicles due to St I started at different time points for each individual vesicle in all lipid compositions reflecting the stochastic nature of the pore-formation process (Figure 3B–D and Appendix A). The permeabilization of the vesicles containing DOPC or DAPC was characterized by faster kinetics in comparison to vesicles containing DMPC, in which permeabilization started slowly without significant initial delay (Figure 3B–E). Moreover, only part of the vesicles containing DAPC were totally permeabilized reaching equilibrium with the external medium (above 80% degree of filling), whereas the vesicles containing DMPC or DOPC reached a different degree of filling under equilibrium (Figure 3B–D).

Two opposite mechanisms of membrane permeabilization have been described: the *all-or-none* mechanism and the *graded* one [18,19,20]. In the *all-or-none* mechanism, once equilibrium is reached the vesicles in the sample exhibit two opposite states: they are either intact or totally permeabilized. Alternatively, in the *graded* mechanism, individual vesicles can exhibit a varying degree of filling at equilibrium (Figure 3I). Figure 3F–H shows the distribution of filling degrees for the individual vesicles in the ensemble of GUVs after 30 min of toxin addition. According to the degree of filling, vesicles were classified into three categories: (*i*) not filled, below 20%; (*ii*) partially filled, (20–80%); or (*iii*) totally filled, above 80%. DMPC:eggSM and DOPC:eggSM vesicles showed a larger heterogeneity in the degree of filling, which in most cases was partial (Figure 3F,H). In contrast, DAPC:eggSM vesicles were either totally permeabilized or remained intact (Figure 3G). This indicates that St I acted through a *graded* mechanism in vesicles with high membrane fluidity (DMPC and DOPC) and obeyed an *all-or-none* mechanism in vesicles with low fluidity (DAPC). From the analysis of the histograms, we calculated the percentage of vesicles that follow a *graded* mode (20–80% degree of filling) or the *all-or-none* mode (above 80% of filling). As expected, the percentage of DMPC:eggSM and DOPC:eggSM vesicles permeabilized via the *graded* mechanism was around 4- and 2.5-fold higher, respectively, than those permeabilized following the *all-or-none* mechanism (Figure 3J). However, the less fluid DAPC-containing vesicles showed a larger proportion of vesicles permeabilized following the *all-or-none* mechanism than vesicles permeabilized through the *graded* one (Figure 3J). These results indicate that the mechanism of membrane permeabilization of St I is tightly modulated by the membrane fluidity of the target membrane.

### 2.4. The Stability of the St I Pore Increases with the Decrease in the Fluidity of the Membrane 

The *all-or-none* mechanism has been associated with the formation of stable pores characterized by a sufficiently long lifetime that allows equilibration between the intra- and extravesicular content. In the *graded* mechanism the pores are supposed to be transient, which does not allow the equilibration of contents [18,19,20]. To investigate whether membrane fluidity impacts the stability of membrane pores formed by St I at equilibrium, we added Alexa 555 after 1-h incubation with Alexa 488, and measured GUVs permeability to both dyes (Figure 4A).

In DMPC and especially in DOPC-containing vesicles, characterized by a high membrane fluidity, most of the GUVs that partially allowed the entrance of Alexa 488 remained impermeable to Alexa 555 (Figure 4B,C,E). However, in the less fluid DAPC:eggSM vesicles, most of the GUVs that were permeable to the first dye also allowed the entrance of the second one to a similar extent (Figure 4B–D). To perform a detailed analysis of the effect of membrane fluidity on the stability of the pores formed by St I, we classified the permeabilized vesicles according to the degree of filling with the first and second dyes. We defined a threshold by which permeabilized GUVs (above 20% of filling) were considered as *gradual* (filling degree of first dye between 20 and 80%) or *all-or-none* (filling degree of the first dye above 80%), and stable (filling degree of the second dye above 80%) or unstable (filling degree of second dye below 80%) (Figure 4F,G). Despite being permeable for both dyes, the majority of GUVs containing DOPC or DMPC were permeabilized in an unstable way following a *graded* mechanism. These results indicate that St I induces transient permeabilization of membranes of high fluidity capable of evolving toward a recovery of its sealed nature at longer times. By contrast, in membranes of low fluidity, pores remain in an open state which allows the complete filling of the GUVs with both dyes. 

### 2.5. The Size of St I Pores Decreases with Membrane Fluidity

A variety of approaches have been used to estimate the pore size of actinoporins by using osmoprotectans of disparate sizes [21,22] and fluorescent dyes [22], or even X-ray crystallography [4] revealing pore radius ranging from 0.6–1.2 nm. To investigate whether the size of the pore formed by St I is related to membrane fluidity, in this work we assessed the permeability of the vesicles to solutes of different sizes. For this, we determined the permeability of the GUVs to free Alexa 555, 3 kDa Dextran-Alexa 488 or 10 kDa Dextran-Alexa 647 upon treatment with St I for 30 min by confocal microscopy (Figure 5A). Approximate hydrodynamic radii for Alexa 555, 3 kDa Dextran Alexa 488, and 10 kDa Dextran 647Alexa were estimated to be: 0.86 nm, 1.4 nm, and 2.4 nm, respectively. As expected, St I followed a *graded* mechanism of permeabilization in vesicles containing DMPC or DOPC characterized by high fluidity and an *all-or-none* mechanism in vesicles of low fluidity (DAPC), without relevant differences between the size markers (Figure 5B–D). From the histograms we calculated the mean degree of filling of the permeabilized vesicles (filling degree above 20%) for each fluorescent marker. This analysis revealed that the vesicles containing DAPC were more permeable to the three solutes in comparison with vesicles containing DMPC or DOPC, which were similar between them (Figure 5E). This result suggests that the pores formed by St I are tunable in size and can be modulated by membrane fluidity. Therefore, in vesicles of high fluidity, St I forms pores of lower size than in low fluidity membranes.

## 3. Discussion

The action mechanism of actinoporins has been largely studied due to their biomedical interest [23,24,25,26]. These toxins are produced as soluble proteins that eventually bind to the lipid membrane where they undergo conformational changes leading to pore formation. This initial step strongly influences their activity in different membrane environments since it determines the protein population that can ultimately form membrane pores. Membrane properties such as lipid composition [27,28], lipid phase coexistence [12,13,14], bilayer thickness [11], and membrane fluidity [10,29] have an impact on the mechanism of how actinoporins bind and permeabilize membranes. Here, we characterized how membrane fluidity influences the mode in which these proteins exert membrane permeabilization. In particular, we investigated the ability of St I to form functional pores in membrane mimetic systems characterized by different fluidity in both bulk and individual vesicles, formed by PC of different acyl chain lengths and unsaturation.

In our work, the processes leading to pore-forming ability of St I in SUVs of POPC and eggSM was maximum in the equimolar mixture and decreases when one of the two components predominated, in agreement with previous studies [10,14,29,30]. The weak affinity and pore-forming ability of St I in SUVs containing a low amount of eggSM has been explained based on different factors [3,29,31]. On the one hand, the lack of activity of St I in pure POPC vesicles, even though the toxin binds to this lipid, may be a consequence of the incapacity of PC molecules to form H-bonds with the toxin, an ability that had been shown to be essential for sticholysins activity [32,33]. In fact, we previously demonstrated that binding of St I to membranes results from an interplay between the presence of SM and membrane fluidity. Once the membrane has a high availability of SM (>30 mol%) its phase state and rheological properties acquire a major role in the recognition and binding steps of the mechanism of action of sticholysins [10]. On the other hand, phase-coexistence could be another factor that enhances activity in conditions near the equimolarity (30 and 70 mol% of eggSM), given that these mixtures of POPC and SM usually display Lα/Lβ phase coexistence [34]. Knowing that St I binds preferentially to more fluid, SM-containing regions of the membrane [10], we could hypothesize that in the equimolar system the confinement of St I in the Lα phase, characterized by a high fluidity, could be considered an efficient strategy to reduce the diffusion area, facilitate the oligomerization, and therefore pore formation in the membrane. On the contrary, the lower permeabilizing activity of St I in vesicles formed by a larger proportion of eggSM could be associated with the unfavorable oligomerization and N-terminal insertion in the highly ordered, less fluid Lβ phase [34].

The inclusion of PCs with different acyl chain lengths (DMPC, DPPC, DSPC and DAPC) and unsaturation (DOPC) in the PC:eggSM binary mixture strongly modulates membrane fluidity. In this study, DMPC:eggSM and DOPC:eggSM membranes exhibited higher fluidity characterized by the increased lateral movement of their components in comparison with those of the rest of the vesicles. An increase in membrane fluidity promoted the association of St I to SUVs and GUVs and consequently its permeabilizing activity. Most likely, membrane fluidity could be an important factor in the oligomerization step and the penetration of the N-terminal region of the toxin into the lipid membrane, since both processes involve the lateral displacement and rearrangement of lipid components. In this sense, we observed that when binding is not a limiting factor (i.e., at saturating conditions), membrane fluidity also potentiated SUVs permeabilization. This indicates that the fluidity of the membrane is a decisive condition for the effectiveness of the post-binding processes (e.g., oligomerization and membrane insertion) that lead to the formation of the lytic structure in the membrane. Another possible explanation to our results could be that changes in the acyl chain length and unsaturation of the PCs would modify bilayer thickness, which in turn could affect actinoporins activity, as was previously demonstrated [11]. According to Lewis and Engelman (1983), bilayer thickness varies linearly with the number of carbons per acyl chain provided these vesicles are in a fluid condition (i.e., the experimental temperature above Tm) [35]. However, this was not the case for the temperature conditions at which our vesicles were in the permeabilization experiments. The only two cases where the experiments were carried out at a temperature within the range of Lβ/Lα phase coexistence were those of DMPC:eggSM or DOPC:eggSM. The other lipid vesicles are below the transition calorimetric range at room temperature and therefore in Lβ phase and accordingly the membrane might show different thicknesses depending on their phase landscape. Taken together, this situation precludes drawing precise conclusions about the influence of membrane thickness on St I activity under our experimental conditions; however, the contribution of bilayer thickness to St I activity cannot be completely ruled out as previously described for sticholysins [11].

The manipulation of membrane fluidity revealed fundamental differences in the nature of the permeabilization process, as shown by single-vesicle analysis in GUVs. In DMPC:eggSM and DOPC:eggSM vesicles, St I followed a *graded* mechanism while liposomes with less fluid bilayers (DAPC:eggSM) underwent an *all-or-none* filling mechanism. A possible hypothesis to explain these differences could be that in vesicles with higher fluidity, the higher efficacy of binding, as well as a favored oligomerization, determines that a high number of vesicles become permeabilized. However, under this condition, the open pore state is also less stable since the high membrane fluidity would favor both formation as well as the dynamic disruption of oligomeric structures. The prevalence of short-living pores in a high number of vesicles could explain why most DMPC:eggSM and DOPC:eggSM GUVs are partially filled. By contrast, in less fluid vesicles (DAPC:eggSM), where the oligomerization process could be disfavored, the formation of oligomeric structures is a relatively infrequent event due to the lower efficacy of the binding and diffusion events, which may explain the larger proportion of GUVs that remain intact for a long time after toxin addition. However, this less deformable environment stabilizes the open state of the fewer pores that are formed by increasing their lifetime. This less frequent, but more effective, event would allow reaching equilibrium with the external medium in those vesicles that are completely permeabilized. This apparent contradiction between the results obtained in SUVs and GUVs could be because in SUVs one single pore could be enough to quickly release the intravesicular content [22,29], therefore the effect of membrane fluidity in the dynamic of pore closing in SUVs is not evident in contrast to what is happening in GUVs. In correspondence with our findings, the study of the permeabilization mechanism of NpreTM, a peptide derived from the membrane-proximal external region of the HIV fusion glycoprotein gp41 subunit, showed that a decrease in membrane fluidity increases the number of vesicles permeabilized trough an *all-or-none* mechanism [20].

The current model of membrane permeabilization by actinoporins propose the formation of small pores of around 2 nm in radius in the membrane [3,4]. Here we show that the stability and size of the pores formed by St I can be modulated by membrane fluidity. In a membrane characterized by high fluidity, the St I pores are smaller and unstable while in vesicles with low fluidity bigger pores remain open for longer time. One possible explanation might be that in membranes with low fluidity characterized by a less deformable environment the pores are more stable on time allowing the entrance of molecules of bigger sizes. By contrast, in membranes with high fluidity the oligomeric structures are less stable oscillating between open and close states, which can explain the partial filling of the vesicles to solutes of different sizes. In this regard, previous studies demonstrated that the actinoporin Eqt II exists in the plasma membrane as a mixture of oligomeric species mostly including monomers, dimers, tetramers, and hexamers but coexisting with a smaller fraction of higher oligomers, most likely octamers and decamers [7]. Interestingly, oligomers of different stoichiometry simultaneously could contribute to membrane permeabilization. Mathematical modeling based on these data supported a new model in which toxin clustering happened in seconds and proceeded via condensation of EqtII dimer units formed upon monomer association; moreover, this modeling also suggested that tetramers were the most stable form of EqtII in the membrane [7]. Given that *i.* tetramers seem to be the most stable ensemble in actinoporins’ membranes, and assuming that *ii.* larger oligomers might originate larger pores, it is tempting to speculate that St II tetramers would be present in both fluidity conditions, but larger oligomers could be stabilized and coexist leading to a fraction of bigger pores only under a less fluid microenvironment.

In this work, it was demonstrated that membrane fluidity, derived from membrane lipid composition, determines the affinity for lipids, the stability and the size of the pores formed by St I. In conditions in which the bilayer is in a fluid state, lipid binding is enhanced, and a high fraction of vesicles are permeabilized. However, in this context most vesicles are just partially permeabilized probably due to a highly dynamic formation/disruption of the pore. On the contrary, when the membrane is less fluid, binding to lipids could be less efficient but pore stability and their size increase. Although being less frequent, because of lower binding-oligomerization events, this mechanism should be more effective due to the enhanced stability of the pores. Probably in living cells, pores with different stability and size coexist in regions of the plasma membrane with different membrane fluidity. In this regard, St I could bind with high affinity to cellular membrane regions with high fluidity forming transient short-lived time pores that might not be enough to produce cell lysis per se, but could activate intracellular processes leading to cell signaling events such as activation of MAPK signaling. Activation of MAPK signaling can lead to cell death as demonstrated for Raji cells, a human B lymphoblastoid cell line [36]. In contrast, in non-tumoral cells (BHK) this activation elicited the mechanism of membrane cell repair [37]. On the contrary, a lower amount of St I could bind with poor affinity to plasma membrane regions of low fluidity resulting in the formation of stable and bigger pores that lead to ion imbalance, and subsequent cell lysis. The cellular impact of each of the two permeabilization processes will probably depend on the state of the cell and the particular properties of its membrane, which would bias the final outcome of the actinoporin action.

## 4. Materials and Methods

### 4.1. Chemicals and Reagents

St I was purified from the sea anemone *S. helianthus* as previously described [1]. The lipids: 1-palmitoyl-2-oleoyl phosphatidylcholine (POPC); egg sphingomyelin (eggSM); 1,2-dimyristoyl phosphatidylcholine (DMPC); 1,2-dipalmitoyl phosphatidylcholine (DPPC); 1,2-distearoyl phosphatidylcholine (DSPC); 1,2-diarachidoyl phosphatidylcholine (DAPC); 1,2-dioleoyl phosphatidylcholine (DOPC) were purchased from Avanti Polar Lipids (Alabaster, AL, USA). The fluorescent probes carboxyfluorescein (CF), 1,1′-Dioctadecyl-3,3,3′,3′-tetramethylindodicarbocyanine perchlorate (DiD), and Alexa 488, Alexa 555 C5 maleimide, Dextran 3000-Alexa 488 and Dextran 10000-Alexa 647 were obtained from Invitrogen (Eugene, OR, USA). Solvents and chemicals were of the highest commercial purity available.

### 4.2. Preparation of Lipid Vesicles

SUVs. The appropriate amounts of lipids dissolved in chloroform:methanol (2:1, *v*:*v*) were mixed and evaporated using a rotoevaporator Bücher 461 (Bücher, Switzerland) for not less than 2 h. Multilamellar vesicles (MLVs) were prepared by hydration of the dried lipid film with TBS (300 mM NaCl, 10 mM Tris-HCl, pH 7.4) for binding assays or with CF (80 mM, pH 7.4) for permeabilizing experiments. MLVs were subjected to six cycles of freezing at −80 °C and thawing at 50 °C. Thereafter, SUVs were prepared by sonication of the MLVs using an ultrasonicator Branson 450 (Branson Ultrasonics Corporation, Danbury, CT, USA.) equipped with a titanium tip. Sonication was performed in 30 cycles of 30 s with resting intervals of 30 s in between. The unstrapped CF was removed by filtering the vesicles through Sephadex G-50 (medium) with TBS. Phospholipid concentration was measured by determining inorganic phosphate according to Rouser’s method [38]. 

GUVs. Vesicles were prepared by the electroformation procedure [39]. A total of 3 µL of the corresponding lipid mixture (2 mg.mL^−1^) containing 0.1 mol% DiD or DiI in chloroform:methanol (2:1, *v*:*v*) were deposited on the platinum wires of the electroformation chamber. After solvent evaporation, the wires were immersed in 300 µL of sucrose (300 mM) and electroformation proceeded for 2 h (10 Hz, 1.4 V) at 65 °C followed by 45 min at 2 Hz, 1.4 V to ensure proper detachment of GUVs from the wires.

### 4.3. Binding of St I to SUVs

Binding of St I to SUVs was followed by the increase in Trp fluorescence. The samples were excited at 295 nm and the emission spectra were recorded from 300 nm to 450 nm. To monitor binding, increasing amounts of SUVs were added to a St I solution (1.5 µM) in TBS and its emission was recorded for each lipid concentration. Fluorescence measurements were carried out at room temperature (23 ± 2 °C) in a spectrofluorimeter Shimadzu RF-5301 PC (Shimadzu, Japan) using 1 cm path length quartz cuvette and excitation and emission slits of 5 nm. The ratio of maximum fluorescence intensity of St I in the presence (F_SUVs_) or absence of SUV (F_0_) was plotted as a function of lipid concentration, and the experimental data were fitted with the Boltzmann equation using Origin 9.0 (Origin Lab Corporation, Northampton, MA, USA) as follows:(1)$$\frac{F_{\mathrm{SUVs}}}{F_{0}}=\frac{-{\left[\frac{F_{\mathrm{SUVs}}}{F_{0}}\right]}_{\max}}{1-e^{\frac{L-L_{50\%}}{dL}}}+{\left[\frac{F_{\mathrm{SUVs}}}{F_{0}}\right]}_{\max}$$
where [F_SUVs_/F_0_]_max_ is the ratio of F_SUVs_/F_0_ obtained under binding saturation conditions at high lipid concentrations, L is the lipid/protein molar ratio and L_50%_ is the lipid/protein molar ratio required for binding 50% of protein molecules present in the assay.

Quenching of Trp fluorescence was achieved by adding the water-soluble quencher acrylamide to a St I solution (1.5 µM) in the presence or the absence of SUVs. Spectral corrections were made by subtracting those spectra collected under identical conditions but without the toxin. The ratio of maximum fluorescence intensities without (F_AA0_) and with the quencher (F_AA_) was plotted as a function of acrylamide concentration ([AA]). The experimental data were analyzed according to the Stern–Volmer equation as follows: (2)$$\frac{F_{\mathrm{AA}0}}{F_{\mathrm{AA}}}=1+{\;K}_{\mathrm{SV}}\left[\mathrm{AA}\right]$$

The slopes of the best-fit linear plots were used to determine the Stern–Volmer quenching constants (K_SV_) [40]. 

### 4.4. Leakage Studies of CF-Containing SUVs

SUVs permeabilization was assessed by measuring the fluorescence (λexc = 490 nm and λem = 520 nm) of CF released induced by the toxin, using a FLUOstar OPTIMA microplate reader (BMG Labtech, Germany). After mixing vesicles and St I, the release of CF produced an increase in fluorescence (f) due to the dequenching of the dye into the external medium, which was resolved in time. Spontaneous (toxin-free) leakage of dye was negligible under these conditions. Total CF release (F_total_) was obtained by adding 1 mM Triton X-100 (final concentration). The fraction of fluorophore release (R_%_) was calculated as follows:(3)$$R_{\%}=\frac{F_{t}-F_{0}}{F_{\mathrm{total}}-F_{0}}\times 100$$
where F_t_ and F_0_ represent the value of fluorescence obtained at time t or before toxin addition, respectively. 

The parameter C_50%_ was calculated by fitting the dose-dependence data at 10 min, with a Hill sigmoid function using Origin 9.0 (Origin Lab Corporation, USA):(4)$$R_{\%}/R_{\max_{10\;\min}}=\frac{R_{\%}/R_{\max}L^{h}}{C_{50\%}^{h}+{\;L}^{h}}$$
where R_%_ was normalized by dividing by R_max_, which corresponds to the maximum release obtained in the presence of high toxin concentration and L is the lipid/toxin molar ratio. C_50%_ is the lipid/toxin molar ratio required for the release of the 50% of CF entrapped in the vesicles. 

### 4.5. Binding of St I to GUVs

50 µL of GUVs made of PC:eggSM containing DiD (0.1 mol%) were added in a LabTec chamber (Nalge Nunc International, Rochester, NY, USA) containing 200 μL of St I 125C-Alexa in PBS to reach a final concentration of (20 nM) in PBS. LabTec chambers were previously incubated with BSA (10 mg.mL^−1^) during 30 min to prevent the unspecific binding of the protein to the chambers. GUVs were observed at 23 ± 2 °C on a LSM710 confocal fluorescence microscopy using a 1.2 C-Apochromat 40× water immersion objective (Zeiss, Jena, Germany). The excitation light was from an Ar-ion laser (488 nm) and He-Ne laser (633 nm). The binding of the protein to the vesicles was estimated at the surface of the vesicles with the plugin Radial Profile Angle (Fiji, NIH, USA). 

### 4.6. Permeabilization of GUVs

The pore-forming activity of St I (20 nM) was measured by adding free Alexa 488 or Alexa 555 or fluorescent labeled Dextrans (100 nM) in phosphate-buffered saline (PBS) to the vesicles and gently mixing. The experiments were performed in LabTec chambers (Nalge Nunc International, Rochester, NY, USA), previously blocked with bovine serum albumin (10 mg/mL), on a LSM710 microscope with a C-Apochromat 40× water immersion objective using as excitation light Ar-ion (488 nm) and HeNe lasers (543 nm and 633 nm). Images were processed automatically using a software implemented in MATLAB (The MathWorks, Inc., Natick, MA, USA) [41]. The percentage of GUVs filling at 30 min was calculated as follows: (5)$$\mathrm{Permeabilized}\;\mathrm{GUVs}\;\left(\%\right)=\frac{F_{30\;min}^{in}-F_{0}}{F_{30\;min}^{out}-F_{0}}*100$$ where $F_{30\;min}^{in}$ and $F_{30\;min}^{out}$ are the average fluorescence intensities inside and outside GUVs at 30 min, respectively, and $F_{0}$ is the background fluorescence at 30 min. Below the threshold of 20% of filling the GUVs were considered non-permeabilized, and above 80% filling GUVs were considered totally permeabilized as previously suggested [39]. 

### 4.7. Membrane Fluidity Determined by Pyrene Fluorescence

SUVs were prepared as described before at 50 µM final concentration and pyrene was added directly to the vesicles after hydration in the range of 0–3 µM. Steady-state fluorescence of pyrene was measured with a spectrofluorimeter Shimadzu RF-5301 PC (Shimadzu, Kyoto, Japan) at room temperature (23 ± 2 °C). Emission spectra were collected from 350 to 600 nm after excitation at 330 nm, using a 1 cm path length quartz cuvette. Data were quantified by calculating the ratio of excimer (470 nm) to monomer (375 nm) fluorescence intensity and plotted as a function of pyrene concentration [17].

### 4.8. Differential Scanning Calorimetric (DSC) Measurements

Calorimetric scans were carried out on a differential scanning calorimeter (MicroCal VP-DSC, Malvern Panalytical Ltd., Malvern, UK) using the software supplied by the manufacturer for data collection and analysis. MLVs were prepared as described above but lipid films were hydrated with 20 mM PIPES buffer (1 mM EDTA, 150 mM NaCl, 0.002% NaN_3_, pH 7.4). The scan rate was 1 °C min^−1^ with a delay of 10 min between sequential measurements in a series to allow thermal equilibration, using PIPES buffer as the reference. The excess heat capacity of the samples was obtained by subtraction of the reference scan of the buffer sample. The temperature of transition (Tm) was determined as the maximum excess heat capacity. Gaussian two-peak fitting analysis with Origin 9.0 (Origin Lab Corporation, Northampton, MA, USA) was performed to deconvolute the thermograms obtained.

## Figures and Tables

**Figure 1 toxins-15-00080-f001:**
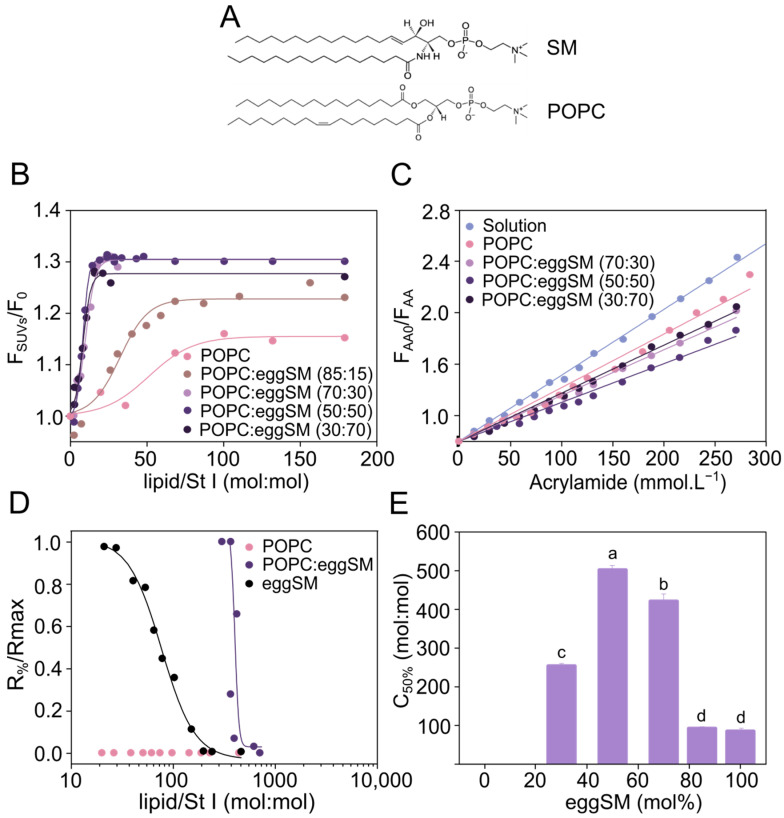
Association and Pore-forming activity of St I in POPC:eggSM vesicles is higher at the equimolar proportion of both phospholipids. (**A**) Chemical structure of SM and POPC. In the case of SM, the shown structure is the most abundant species in the natural eggSM lipid mixture. (**B**) Increase in the intrinsic protein fluorescence intensity as a function of the lipid/St I molar ratio. Fluorescence intensities at a given lipid concentration (F_SUVs_) and in the absence of vesicles (F_0_) at 332 nm. The figure shows a representative result from three independent experiments. Lines are the best fit of the experimental data to a Boltzmann function (R2 > 0.97). (**C**) Acrylamide quenching of the intrinsic fluorescence of St I in solution and upon binding to vesicles. Fluorescence intensities were measured in the absence (F_AA0_) and the presence of acrylamide (F_AA_) at 332 nm. Lines are plotted according to the Stern–Volmer equation (r2 > 0.98). St I concentration: 1.5 µmol.L^−1^, λ excitation: 295 nm, and λ emission: 300–450 nm. (**D**) Increase in CF release from SUVs after 10 min of toxin addition. The maximal release obtained at high toxin doses (R_max_) was slightly different among different vesicle compositions. Therefore, the reported release values (R_%_) were normalized by dividing by R_max_. Solid lines are the best fit of the experimental data to a Hill function (R > 0.8). (**E**) Effect of eggSM molar fraction on the vesicles in the permeabilizing activity of St I, expressed as the Lipid/St I molar ratio required to cause lysis of 50% of liposomes in the assay (C_50%_). The higher the C_50%_ the lower the amount of protein required to permeabilize 50% of the liposomes, reflecting a condition of higher St I activity. Statistical analysis was performed with one-way ANOVA with Tukey as post hoc test. Letters a, b, c, and d indicate independent groups with significant differences among them (*p* < 0.05).

**Figure 2 toxins-15-00080-f002:**
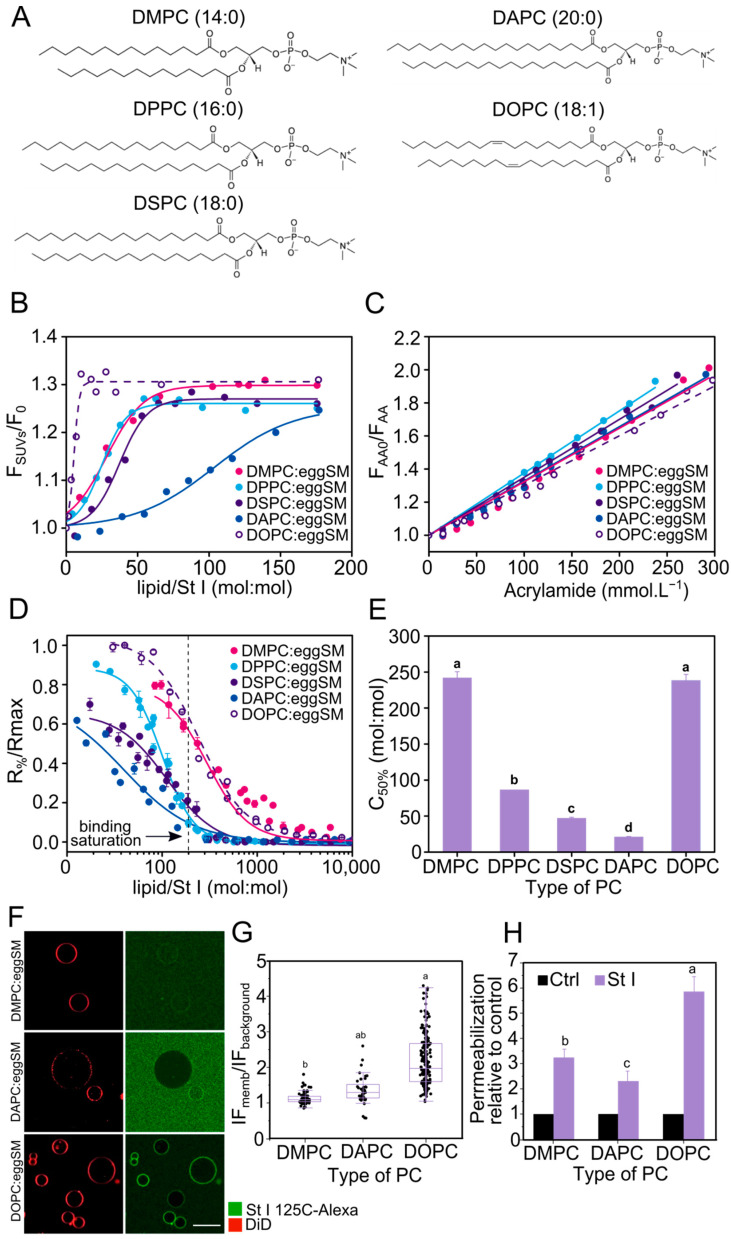
St I binds and permeabilizes with higher efficiency liposomes of PC:eggSM with high membrane fluidity. (**A**) Chemical structure of DMPC, DPPC, DSPC, DAPC and DOPC. (**B**) Increase in the intrinsic protein fluorescence intensity at 332 nm as a function of the lipid/St I molar ratio. Fluorescence intensities at a given lipid concentration (F_SUVs_) and in the absence of vesicles (F_0_). The figure shows a representative result from three independent experiments. Lines are the best fit of the experimental data to a Boltzmann function (R2 > 0.98). (**C**) Fluorescence intensities measured in the absence (F_AA0_) and in the presence of acrylamide (F_AA_) at 332 nm. Solid lines are plotted according to the Stern–Volmer equation (r2 > 0.97). St I concentration: 1.5 µmol.L^−1^, λ excitation: 295 nm, and λ emission: 300–450 nm. (**D**) Dose-dependence of CF release induced by St I after 10 min of toxin addition. Points are the mean value, and bars indicate the standard deviation from a set of three independent experiments. When no error bar is observed the corresponding standard deviation is smaller than the size of the symbol. Lines are the best fit of the experimental data to a Hill function (R2 > 0.97). The vertical dotted line indicates lipid/St I molar ratio (200:1), at which saturation of the binding was reached for all lipid compositions. (**E**) Effect of PC acyl chain length and unsaturation in equimolar PC:eggSM mixtures on the permeabilizing activity of St I expressed as C_50%_. (**F**) Confocal microscopy images of GUVs of DMPC:eggSM (50:50), DAPC:eggSM (50:50) and DOPC:eggSM (50:50) containing 0.1% DiD after 30 min of incubation with St I 125C-Alexa (20 nM), Scale bar: 20 µm. (**G**) Effect of PC acyl chain length and unsaturation on the binding of St I 125C-Alexa to PC:eggSM (50:50) GUVs. Each dot represents one GUV. At least 50 vesicles were analyzed in each composition. (**H**) Permeabilizing activity of St I in GUVs of PC:eggSM (50:50) containing different PCs. The pore-forming activity was calculated as the percentage of filled vesicles 30 min after addition of St I (20 nM) and normalized by dividing by the percentage of permeabilized vesicles in the absence of St I (Ctrl). Below a threshold of 20% of filling, the GUVs are considered non-permeabilized. In each experiment, we used GUVs preparations in which the spontaneous permeabilization was lower than 15% of the analyzed vesicles in the timeframe of the experiment. Bars show the mean and SD from a set of at least three independent experiments. Between 400 and 1800 vesicles were analyzed for each composition. Statistical analysis was performed with one-way ANOVA with Tukey as post hoc test. The letters a, b, c and d indicate independent groups with significant differences among them (*p* < 0.01).

**Figure 3 toxins-15-00080-f003:**
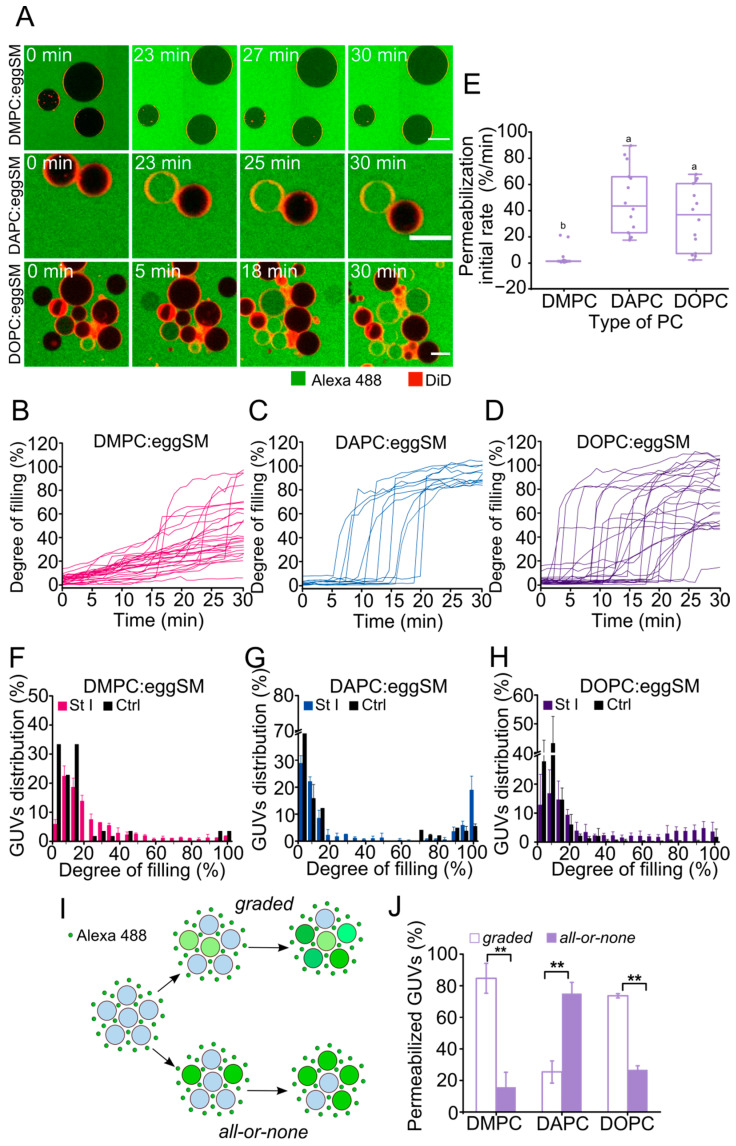
The mechanism of vesicle permeabilization mediated by St I is regulated by membrane fluidity. (**A**) Time-lapse confocal images of DMPC:eggSM (50:50), DAPC:eggSM (50:50) and DOPC:eggSM (50:50) of GUVs containing 0.1% DiD after incubation with St I (20 nM) and free Alexa 488. Scale bar: 15 µm. (**B**–**D**) Kinetics of single-vesicle permeabilization by St I of GUVs of different compositions. (**E**) Effect of the PC on the initial rate of vesicle permeabilization. This parameter was calculated from single-vesicles kinetics similar to the ones shown in (**B**–**D**). Each dot represents one individual vesicle. At least 14 vesicles were analyzed for each lipid composition. (**F**–**H**) Distribution of the GUVs as a function of the filling degree after 30 min of incubation with or without St I (Ctrl). Bin size 5%. Each bar represents the mean and SD of three independent experiments. Between 400 and 1800 vesicles were analyzed for each lipid composition. (**I**) Graphical representation of the permeabilization of individual vesicles following a *graded* or an *all-or-none* mechanism. (**J**) GUVs permeabilized (above 20% of filling) were classified according to their degree of filling with Alexa 488 as *all-or-none* (above 80% of filling) or *graded* (20–80% of filling). Stable refers to those GUVs that reach equilibrium between the intra- and extravesicular medium while unstable stands for GUVs that reach an intermediate degree of filling. In each experiment, we used GUVs preparations in which the spontaneous permeabilization was lower than 15% of the analyzed vesicles in the timeframe of the experiment. Error bars correspond to the SD calculated from three independent experiments. The statistical comparison of the predominant mechanism for each composition was performed using a Student’s *t* test. ** indicates independent groups with significant differences among them (*p* < 0.01).

**Figure 4 toxins-15-00080-f004:**
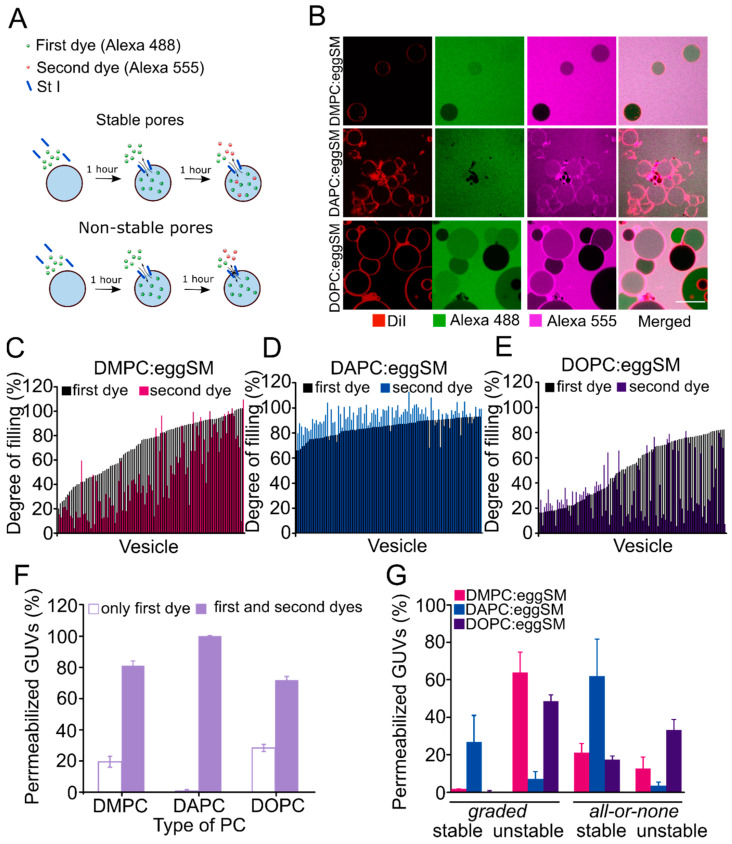
The stability of St I pores is modulated by membrane fluidity. (**A**) Graphical representation of the pore stability experiment. Free Alexa 488 was added simultaneously with St I to the GUVs and free Alexa 555 was added 1 h later. (**B**) Confocal images of GUVs containing 0.1% of DiD after 2 h of incubation with St I (20 nM). Scale bar: 15 µm. (**C**–**E**) Quantification of the filling degree of individual GUVs with the first dye (Alexa 488) and the second dye (Alexa 555) after 2 h of incubation with St I. For each composition, 100 vesicles were measured and organized by the degree of filling with the first dye. (**F**) Percentage of vesicles that are permeable only to the first dye or to both. (**G**) GUVs permeabilized (above 20% of filling) were classified according to their degree of filling with Alexa 488 as *all-or-none* (above 80% of filling) or *graded* (20–80% of filling). Stable refers to those GUVs that reach equilibrium between the intra- and extravesicular medium, while unstable stands for GUVs that reach an intermedia degree of filling. In each experiment, we used GUVs preparations in which the spontaneous permeabilization was lower than 15% of the analyzed vesicles in the timeframe of the experiment. Error bars correspond to the SD calculated from three independent experiments.

**Figure 5 toxins-15-00080-f005:**
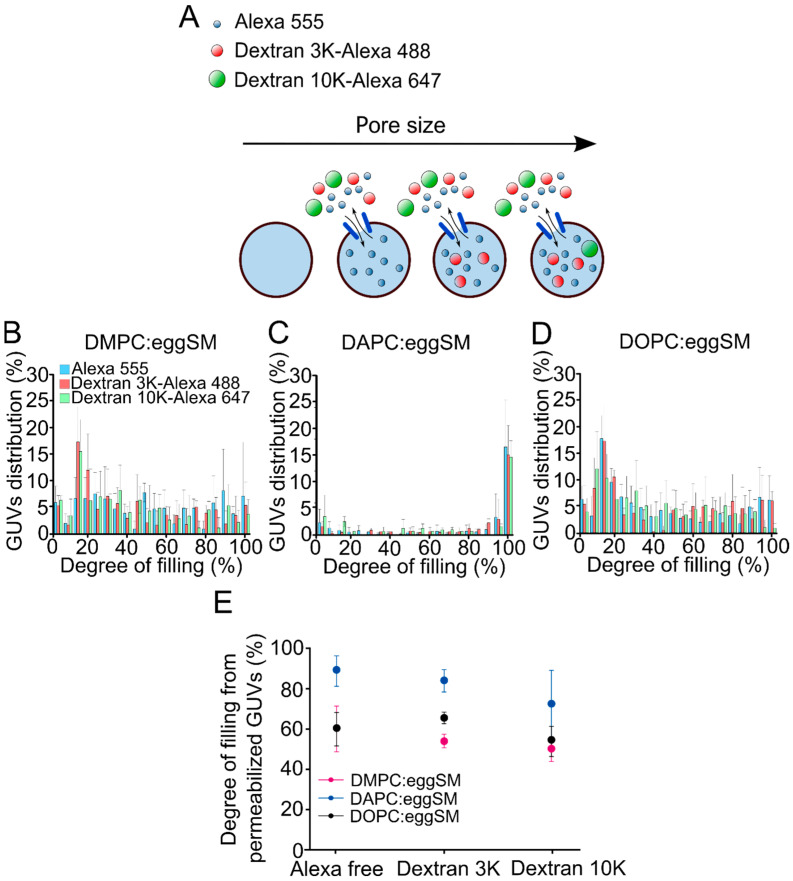
The size of St I pores is modulated by membrane fluidity. (**A**) Graphical representation of the pore size experiment. Free Alexa 555, Dextran 3K-Alexa 488 and Dextran 10K-Alexa 647 were added simultaneously with St I to the GUVs and the degree of vesicle filling was measured after 30 min. (**B**–**D**) Distribution of the GUVs as a function of the filling degree with Alexa 555, Dextran 3K-Alexa 488 or Dextran 10K-Alexa 647 after 30 min of incubation with St I. Bin size 5%. Each bar represents the mean and SD of at least three independent experiments. Between 500 and 1000 vesicles were analyzed for each condition. (**E**) Degree of filling of permeabilized GUVs as a function of the reporter fluorescent dye in the external medium at 30 min. Dots represent the mean of the degree of filling from permeabilized vesicles (above 20% of filling) from at least three independent experiments. In each experiment, we used GUVs preparations in which the spontaneous permeabilization was lower than 15% of the analyzed vesicles in the timeframe of the experiment.

**Table 1 toxins-15-00080-t001:** Parameters characterizing association of St I to SUVs of POPC:eggSM.

Composition	F_SUVs_/F_0 max_	L_50%_ (mol:mol)	Δλ_SUVs-sol_ (nm)	K_SV_ (M^−1^)
Solution	-	-	-	5.1 ± 0.1
POPC	1.15 ± 0.01	51.3 ± 7.7	2 ± 1	4.2 ± 0.1
POPC:eggSM (85:15)	1.23 ± 0.01	33.5 ± 2.7	2 ± 1	-
POPC:eggSM (70:30)	1.30 ± 0.01	9.9 ± 0.5	6 ± 1	3.6 ± 0.1
POPC:eggSM (50:50)	1.30 ± 0.01	8.2 ± 0.2	5 ± 1	3.5 ± 0.1
POPC:eggSM (30:70)	1.27 ± 0.01	8.1 ± 0.4	6 ± 1	3.7 ± 0.1

F_SUVs_/F_0 max_: ratio between the fluorescence intensities of the bound and free fraction of St I at high lipid/toxin molar ratio, calculated from data shown in Figure 1B. L_50%_: lipid/toxin molar ratio necessary to bind half of the population of the protein to the liposomes, calculated from data shown in Figure 1B. Δλ_SUVs-sol_: shift of the maximum emission wavelength in the presence of saturating lipid conditions (lipid/St I molar ratio ~ 200) regarding the protein in solution (332 nm). K_SV_: Stern–Volmer constant obtained by regression analysis of data represented in Figure 1C. F_SUVs_/F_0 max_ and L_50%_ contain the standard error of the experimental fitting to the Boltzmann function of a typical experiment from a series of two independent experiments. Δλ_SUVs-sol_ and K_SV_ are the mean and SD of at least two independent experiments.

**Table 2 toxins-15-00080-t002:** Effect of PC acyl chain length and unsaturation on St I association to PC:eggSM (50:50) SUVs.

Composition	F_SUVs_/F_0 max_	L_50%_ (mol:mol)	Δλ_SUVs-sol_ (nm)	K_SV_ (M^−1^)
DMPC:eggSM	1.30 ± 0.01	28.3 ± 2.4	5 ± 1	3.2 ± 0.1
DPPC:eggSM	1.26 ± 0.01	24.9 ± 1.1	5 ± 1	3.7 ± 0.1
DSPC:eggSM	1.27 ± 0.01	37.6 ± 1.8	3 ± 1	3.5 ± 0.1
DAPC:eggSM	1.25 ± 0.01	103.0 ± 8.0	5 ± 1	3.3 ± 0.1
DOPC:eggSM	1.31 ± 0.01	5.4 ± 0.5	5 ± 1	3.0 ± 0.1

F_SUVs_/F_0 max_: ratio between the fluorescence intensities of the bound forms and free St I at high lipid/toxin molar ratio calculated from data shown in Figure 2B. L_50%_: lipid/toxin molar ratio necessary to bind half of the protein ensemble to the liposomes, calculated from data shown in Figure 2B. Δλ_SUVs-sol_: shift of the maximum emission wavelength in the presence of saturating lipid conditions (Lipid/St I molar ratio ~ 200) regarding the protein in solution (332 nm). K_SV_: Stern–Volmer constant obtained by regression analysis of data represented in Figure 2C. F_SUVs_/F_0 max_ and L_50%_ contain the standard error of the experimental adjustment to the Boltzmann function of a typical experiment from a series of two independent experiments. Δλ_SUVs-sol_ and K_SV_ are the mean and SD of at least two independent experiments.

## Data Availability

Not applicable.

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
