# Peer review of "The Important Role of Membrane Fluidity on the Lytic Mechanism of the α-Pore-Forming Toxin Sticholysin I"

_toxins, 2023, doi:10.3390/toxins15010080_

Round 1

Reviewer 1 Report

The authors describe in their article experiments which indicate that the fluidity of the membrane plays a role not only for the extent of pore formation by sticholysin I, but also has an impact on the size and stability of the pore.

To understand the mechanism of pore formation and modulation of the pore properties is an important aspect in the field in general to understand the pathogenicity of the corresponding organism. Additionally, there is some interest to apply pore forming toxins in medical aspects, for which the properties of the pore have to be known as detailed as possible. Thus, the question addressed in this study is a relevant.

In order to address the role of membrane fluidity, the interaction of StI with artificial membranes was studied: one hand with SUVs, on the other hand with GUVs in order to study the interaction on a single liposome level. While overall the main conclusions drawn are valid, with respect to some details the manuscript should be improved, as listed below:

Major comments

1)     In the whole discussion about the effect of membrane fluidity, binding and pore formation is discussed as separable processes. However, this is not the case. When monitoring binding via changes in Trp fluorescence, the total amount of bound toxin, in all its possible oligomerization states, is observed. It is very likely, that oligomers have a higher affinity (or lower off-rate) than monomers, and in turn, that pores are more or less irreversibly bound. Thus, it is impossible to distinguish between a higher level of membrane bound toxin due to increased binding affinity (on the level of monomer), or due to an increase in oligomerization efficiency. The text should be thoroughly screened to identify the positions, at which the wording should be altered to avoid the suggest to the reader, that the initial binding process can be separately monitored.

2)     In the discussion it is mentioned in the beginning, that presence of SM is important for binding; later on only fluidity is mentioned. The absence of efflux and weak binding in case of POPC shows, that SM has to be present to allow sufficient binding to form pores; this could be more emphasized in the discussion.

3)     It should also be emphasized, that in case of SUVs usually one pore is sufficient to quickly release the content; therefore, in contrast to GUVs, dynamics of pore closing is not relevant for SUVs, except if very fast.

4)     The experiment where a second dye was added after the first one was a clever idea. However, I wonder: if the second dye cannot enter the GUVs within 1 h, does this mean that previously existing pore have vanished to exist? Is really just a difference in dynamics (switching between open and closed on a shorter time scale compare to gel-phase membrane) able to explain the results?

5)     In the discussion, similar experiments on other related toxin should be included, e.g. Palacios-Ortega, Juan, et al. "Regulation of sticholysin II-induced pore formation by lipid bilayer composition, phase state, and interfacial properties." Langmuir 32.14 (2016): 3476-3484.  …but there are surely more.

6)     In experiments with GUVs, DMPC, DAPC and DOPC was compared. Problem with GUVS: difficult to control the lipid concentration; also, the size (and with this curvature) depends on composition. Could you comment on that ?

7)     DSC experiments: if in a mixture of two lipid the TM observed is in between the one observed individually, as observed for DOPC/SM and DMPC/SM mixtures, this is an indication of mixing, not phase separation. Only if two peaks are observed, one can infer that the lipids demix, as with DAPC. In case of DPPC, one cannot tell, since they have a very similar TM. The TM of DSPC is 55°C; thus here also mixing seems to occur. See Schwieger, C., Blume, A. Interaction of poly(l-lysines) with negatively charged membranes: an FT-IR and DSC study. Eur Biophys J 36, 437 (2007). https://doi.org/10.1007/s00249-006-0080-8 for reference of this topic.
Please change the text accordingly.

8)     Experiments of permeabilization in case of GUVs were made with a very low protein/lipid ratio, completely different from the one with SUVs. Thus, one wonders to what extent the results can be related to each other.

Minor comments

-        Abstract, line 9: I would add “SM with saturated fatty acids”, since there are also other (e.g. OSM)

-        Page 2, second paragraph: “St I lipid binding, which anticipates membrane penetration and pore formation”. Not clear, what is meant here. Furthermore, as described above, binding, oligomerization and pore-formation cannot be separated in the binding assay.

-        Same paragraph, last sentences: I think that “precisely regulated” is a too strong wording.

-        Page 4, third paragraph: “to compare the fluid condition ..” please reconsider phrasing.

-        Fig. 1c: which wavelength was used for the plot?

-        Fig. 2d: any idea why some curves seem to fail to approach 100% efflux?

-        Fig. 2g: black points would be clearer than violet ones.

-        At several places: a threshold is a certain value, thus “a threshold of <= 20% does not make sense. One would rather say: below a threshold of 20% of filling, the GUVs are considered as non-permeabilized and so on.

-        Fig.2f: in the red channel, the GUVs are barely visible.

-        Table 2: “the presence of two monounsaturations”. Please rephrase

-        Page 5, first paragraph: “Since under this condition binding equilibrium was reached”. You mean probably something else: under this conditions, all protein is bound, therefore the overall ratio of protein/liposome is very similar,….

-        Page 5, last paragraph: the finding, that in the GUV assay you see a difference with respect to permeabilization, but not with SUVs, should be explained. Possibly, the temperature was  bit higher in the GUV measurement (considering, that the Tm of DMPC is close to the working temperature). A figure like Fig. 3b-d should be provided for the controls in the supplements.

-        Fig. 3 e: not clear, what the rates are.

-        Fig3J: how do the controls look like?

-        Page 7, first paragraph, last sentence: please rephrase, the sentence is not clear.

-        Fig.4 C-E: label of the x-axis is missing.

-        Fig.4, G: please explain a bit more.

-        Page 10, end of page: the approx.. diameter of the three dyes should be provided

-        Fig. 5E: I have difficulties to reconcile figures B-D with the results shown in E. Are these really the mean values of the data shown above?

-        Page 13: second paragraph: “However, formation of tetramers is faster than..” Kinetics of formation is not related to stability in an unambiguous way. Please rephrase or omit.

-        Page 13, paragraph just before experimental procedures: what is “consequent lysis”?

-        Page 14, paragraph 4.2: presumably, the CF solution was also buffered. How did you make sure, that no osmotic pressure due to different osmolarities inside/outside the liposomes occur?

-        Paragraph 4.3.
1) I sincerely hope, that the data were not adjusted to anything. …proper phrasing:
the data were fitted with a Boltzmann-equation as given…
2) something is wrong with the equation: at L=0 it should yield 1, which it does not; at L=L50 the denominator is not defined. Please check your fits.

-        Paragraph 4.4: how did you make up your mixture of liposome and toxin? If a small volume of highly concentrated toxin solution is added, the results are usually unreliable, since before proper mixing already pores have formed, so results do not reflect the final toxin concentration properly.

-        Paragraph 4.5: “50 ul” should be corrected

-        Paragraph 4.6: was alexa555 also irradiated with the 488 nm laser?
Here, a threshold of 85% is given, in the main text it is 80%...

-        Why where the DSC experiments performed in a different buffer?

-        Statistics: two experiments do not allow to calculated a SD. Here, one should just report the results of the two measurements

Reviewer 2 Report

The present paper is a study that evaluates the relevance of membrane fluidity on the lytic mechanism of sticholysin I (St I), an α-pore-forming toxins produced by the sea anemone Stichodactyla helianthus. Specifically, they are interested in how the biophysical properties of the membrane modulate the properties of the pores formed by actinoporins. The authors use different membrane composition to study the membrane binding capability of St I, as well as its ability to form pores. They register the changes in tryptophan emission and its susceptibility to acrylamide. To test the permeabilizing activity of St I, they monitor the release of carboxyfluorescein from SUVs and the entry of different probes into GUVs.

The outline of the paper is very good. However, some of their conclusions are not clearly justified, or do not rule out other interpretations that could be mentioned. I think the authors should address and/or comment some key points and questions listed below. I recommend major revisions.

Major concerns:

- To change membrane fluidity, the authors use DMPC, DPPC, DSPC, and DAPC, and DOPC. As the authors mention, Palacios-Ortega et al. (ref. 11 of the manuscript) showed that bilayer thickness affects actinoporins. The listed lipids not only produce membranes of different fluidities, but also of different thicknesses. The authors should clarify if bilayer thickness affects their results and how.

- The authors use, all at the same temperature, some membranes with Lα/Lβ coexistence, and others in the Lβ phase. In the DAPC:egg SM mixture they observe DAPC-enriched and egg SM-enriched phase coexistence. The absence of two transition temperatures in the other Lβ mixtures could imply a better PC:SM mixing. This should be clarified, and the authors should address the possible relevance of this effect in their results.

- To get better insight on the membrane fluidity of the membranes, the authors could add measurements of DPH/TMA-DPH anisotropy.

- Using R%/Rmax results in a loss of information. What is the final release observed in the carboxyfluorescein-SUV experiments relative to total release induced by Triton X-100? This is particularly relevant as graded and all-or-none mechanisms are later discussed regarding GUV experiments. Release after 10 min from DAPC-containing SUVs is ~60% of DOPC-containing vesicles in saturation conditions. Different mechanism as in GUVs? Please explain.

- Section 2.4. The authors see that Alexa-555 cannot enter the GUVs containing DMPC or DOPC after 1 h incubation, as if the pores were sealed. This is not the case for DPAC GUVs, since they observe entry of the fluorophore. Palacios-Ortega and coworkers (2020, BBA-Biomembranes) showed, using fluid membranes of DOPC:eSM:Chol, that St II pores were sealed for calcein, but still open at that point, as they allowed for dithionite influx. They explained these observations saying that perturbations on the membrane during the first stages of the pore formation process would allow for the passage of calcein and similar fluorophores, that would be too large for the stable pores. This explanation could as well justify the authors’ observations on GUVs with DMPC and DOPC. Due to St I’s low affinity for DAPC:egg SM bilayers, it would take longer to form stable pores. As shown in figure 2F, labeled St I remain can remain unbound for longer times. Together with the above explanation, retarded St I binding could also explain permeability for Alexa 555. Also, longer lived imperfections cause by membrane rigidity of DAPC:egg SM membranes could also explained the observed all-or-none effect on those membranes, and the permeability for larger molecules. Following this reasoning, different pore sizes are not required either to explain these observations.

- The authors write “…tetramers seem to be the most stable ensemble in actinoporins’ membranes… larger oligomers might originate larger pores”. Please justify the claim on tetramers and hypothesize how a relatively fixed monomer geometry can give rise to such varied and geometrically different stoichiometries starting as low as tetramers.

- The authors write “…the open pore state is also less stable…”, “…short-lived time pores that might not be enough to produce cell lysis but could activate intracellular processes leading to different cell signaling events… membrane-repair damage mechanism.” What would be the natural advantage of a pore that closes, and would thus be “used” mostly to “just” activate cellular defensive processes? In the abovementioned paper, Palacios-Ortega et al. showed that pores on fluid membranes are permeable to dithionite, and hence presumably to cellular monoatomic ions, enough to kill the cell.

Minor concerns

- The authors use “… a fluorescently labeled St I (St I-Alexa 488).” They should indicate the mutant used, if so. Does labeling at 125C affect the behavior of St I? In that paragraph, they imply that St I-Alexa 488 is used together with free Alexa 488. A different fluorophore should have been chosen in that case to differentiate the molecules. If not the case, please rephrase.

- I could have missed it, but the authors should clearly show/state if intact GUVs’ permeability is altered in the timeframe of the experiments.

Other comments

- In the introduction, it is said that actinoporins are basic proteins with “high affinity for SM-containing membranes”, implying that the presence of SM in the membrane is enough for actinoporin-binding. However, Alm and coworkers (2015, BBA-Biomembranes) showed that the addition of a saturated ceramide greatly diminished the affinity of St II for those membranes despite their content of SM. Please rephrase.

- A few lines later, “five sites for membrane binding…two sites for lipids with other headgroups…”. In Tanaka et al. (Nat. Comms. 2015), five sites for “lipid” binding are proposed. These would be used for membrane binding. Phrasing can be interpreted as if they could act separately. However, Tanaka et al. speak of “lipid multivalency” that “may increase the affinity of the toxin for the membrane”, also postulating two of the sites are “low-affinity sites, or perhaps high-affinity binding sites for lipids with headgroups other than POC”. The authors’ claim is not justified by the provided reference. Please add a new appropriate reference and rephase.

- In the reagents section, the authors clarify that they use egg SM. However, along the paper they only indicate SM. The specific SM used should be specified more clearly along the paper, as its structural properties are determining the membrane’s biophysical state. Please see Arsov et al. 2018, Chem. Phys. Lipids.

- The authors state “…vesicles (with) weaker lipid cohesion provide a more suitable environment for St I lipid binding, which anticipates membrane penetration…”, implying a clear distinction between binding and membrane penetration. However, these two processes are probably coupled in an equilibrium. Easier pore formation will probably displace the equilibrium. Weaker lipid cohesion would therefore facilitate binding, pore formation, both directly, one through the other?

- The authors write “…be precisely regulated by the fluidity of the different regions of the membrane.” However, the pores are completely foreign to the attacked cell, and would not regulate that. Please rephrase.

- Figure 1a. “Chemical structure of SM and POPC.” Egg SM is not homogenous. Please indicate that the shown structure is the most abundant. If possible, remove the frames around the molecular structures.

- Figure 2. a) The structures are too small, should be larger to appreciate the differences. Remove frames. d) Also, please justify using Hill and Boltzmann’s functions for fitting in the procedures section.

- Section 2.2, first line, “PC:SM”. The authors probably mean POPC:egg SM. Please clarify.

- The procedures followed for the pyrene experiments should be clarified. Was the pyrene added before or after hydration? If before hydration, the molar fraction relative to the lipid mixture should be stated. Some emission spectra of pyrene could be shown as supplementary, as well as some DSC traces.

- In “…any difference between these latters them.” “them” should be removed.

- In “…differences in pore-forming stages after to the binding step.” “to” should be removed.

- In the title of table 2, “chain” is likely missing an “s”.

- In “…mechanism of membrane permeabilization of St I is tightly regulated by the membrane.” Same as before. “Dependent” may be a better choice than “regulated”.

- Figure 3, panels B, C, and D. These clearly show the different timing of the process. Maybe the authors could consider making a separate graph, alike to panel E, in which they quantify these delays, and change panels B, C, and D, synchronizing the starts, to better appreciate the reproducibility or divergence of the traces upon start. Panel J shows some small rectangles that probably shouldn’t be there. Please check in case it’s just the pdf copy.

- Relative to section 2.5, the authors should comment on previous studies on pore size, pore diameter, and size of solutes allowed through.

- Figure 5. Colors in panel E are hard to distinguish. Please modify. A) Remove the red circle in the second vesicle of the scheme. If increasing pore size discriminates size of solutes that pass, according to the authors’ model, it would discriminate size, not amount.

- Ref. 29 probably should be Maula et al. 2013, BBA-Biomembranes instead?

- Reagents: Why use Alexa 555 C5 maleimide?

- Was the size of the SUVs used checked?

- Eq. 3 should include “x 100”, otherwise it’s not a percentage.

- Section 4.5. Change “50 uL” by 50 µL, remove space in “200 µ L”, change “EUA” for USA.

- Ref. 8 has a typo in “MačEk”.

- Please use italics in the scientific names. Correct at least in refs. 13, 25, and 27.

Round 2

Reviewer 2 Report

I would like to thank the authors for a nice paper, nice answers to my questions, and a valuable discussion.

While I don't fully agree with some of their conclusions, their results are valuable, and their discussion provides interesting points of view and starting points for further work to clarify possible disagreements.

I recommend acceptance after minor revisions. Some minor comments:

- Please add a phrase regarding the possible effect of bilayer thickness (unless I missed it in the main text). Simply something similar as what was written in the reply would be nice to make the reader aware that this possibility is not completely ruled out.

- I think Figures R1, R2, and R3, and Table R1 could be added as supplementary material. These are nice figures, and can be handy for the reader.

- Addition of pyrene after hydration should be mentioned in the main text.

I hope the authors found my criticism constructive and helpful.

Best regards, and Happy New Year!
